# Poor coherence in older people's speech is explained by impaired semantic and executive processes

**Paul Hoffman\*, Ekaterina Loginova, Asatta Russell**

Centre for Cognitive Ageing and Cognitive Epidemiology (CCACE), Department of Psychology, University of Edinburgh, Edinburgh, United Kingdom

**Abstract** The ability to speak coherently is essential for effective communication but declines with age: older people more frequently produce tangential, off-topic speech. The cognitive factors underpinning this decline are poorly understood. We predicted that maintaining coherence relies on effective regulation of activated semantic knowledge about the world, and particularly on the selection of currently relevant semantic representations to drive speech production. To test this, we collected 840 speech samples along with measures of executive and semantic ability from 60 young and older adults, using a novel computational method to quantify coherence. Semantic selection ability predicted coherence, as did level of semantic knowledge and a measure of domain-general executive ability. These factors fully accounted for the age-related coherence deficit. Our results indicate that maintaining coherence in speech becomes more challenging as people age because they accumulate more knowledge but are less able to effectively regulate how it is activated and used.

DOI: https://doi.org/10.7554/eLife.38907.001

**\*For correspondence:**
p.hoffman@ed.ac.uk

**Competing interests:** The authors declare that no competing interests exist.

## Introduction

Engaging in a conversation is a complex cognitive act, in which a speaker must settle on a topic for discussion, generate a series of appropriate, relevant and hopefully interesting statements and monitor their speech as the discourse unfolds to ensure that they remain on-topic. Discourse that successfully navigates these challenges is said to be coherent: it consists of a series of well-connected statements all related to a shared topic, making it easy to comprehend (*Foltz, 2007*; *Glosser and Deser, 1992*). The ability to produce coherent speech tends to decline as people grow older. Older adults are more likely to produce tangential, off-topic utterances in conversation (*Arbuckle and Gold, 1993*; *Glosser and Deser, 1992*) and to provide irrelevant information when telling a story (*Juncos-Rabadán et al., 2005*; *Marini et al., 2005*) or describing an object (*Long et al., 2018*). Such changes reduce the effectiveness of communication and the quality of older people's verbal interactions (*Arbuckle et al., 2000*; *Pushkar et al., 2000*). Indeed, less coherent speech is associated with higher levels of stress and less satisfaction in social interactions (*Arbuckle and Gold, 1993*; *Gold et al., 1988*; *Pushkar et al., 2000*). Researchers have often made a distinction between local coherence (LC), the degree to which adjoining utterances relate meaningfully to one another, and global coherence (GC), the degree to which each utterance relates to the topic under discussion (*Glosser and Deser, 1992*; *Kintsch and van Dijk, 1978*). Most studies have reported larger declines in GC in later life, though reductions in LC have also been observed (*Glosser and Deser, 1992*; *Kemper et al., 2010*; *Marini et al., 2005*; *Wright et al., 2014*).

Explanations for age-related decline in coherence have typically focused on deterioration in domain-general cognitive control processes, which are assumed to be involved in the monitoring and selection of topics during speech (*Kintz et al., 2016*). This view is supported by a handful of

**eLife digest** During a conversation, each person must plan and monitor what they say to make sure it is relevant to the discussion. This is called maintaining coherence during speech. Studies suggest that as people get older they find it harder to remain coherent, and become more likely to produce irrelevant or off-topic information when speaking. This reduces how effectively they communicate and can have negative effects on their social interactions. However, little is known about how thinking skills affect coherence in speech and why this declines in later life.

To investigate, Hoffman et al. asked two groups of volunteers – a 'younger' group made up of people aged between 18 and 30 years old, and an 'older' group of people aged over 60 – to perform various speech-related tasks. For example, the volunteers were asked to speak when prompted and a computer analysis was used to evaluate how coherent they were. They also completed a speech test while distracted, and took part in tests to understand how well they can suppress irrelevant information.

The results of the tests show that three factors influence how coherent people are during conversations: how well they could control and regulate their behaviour, how much general knowledge they had, and how skilled they were at selecting the most relevant information for the task they were doing.

Having larger stores of knowledge to select from increases the challenge of staying on topic for older people. At the same time, they may experience age-related declines in the ability to suppress unnecessary information. This may help to explain why some people become less coherent as they get older and why some do not.

DOI: https://doi.org/10.7554/eLife.38907.002

studies reporting that coherence in speech is predicted by performance on non-verbal tasks requiring cognitive control (*Gold et al., 1988*; *Kintz et al., 2016*; *North et al., 1986*). One particular theory holds declines in coherence result from a reduced ability to inhibit irrelevant information, which means that older people are less able to prevent irrelevant or off-topic ideas from intruding into their discourse (*Arbuckle and Gold, 1993*; *Marini and Andreetta, 2016*). This view is supported by evidence that performance on cognitive tests requiring inhibition, such as the Stroop test and Trails test, predicts the level of coherence in older adults' speech (*Arbuckle and Gold, 1993*; *Wright et al., 2014*). However, this factor does not appear to provide a complete explanation for age-related coherence declines. This is likely because previous studies have overlooked the critical role that conceptual knowledge about the world (i.e., semantic knowledge) plays in the generation of meaningful, coherent speech. Here, we propose and test the hypothesis that coherence depends not only on domain-general executive resources, but specifically on the ability to regulate the activation of semantic knowledge, ensuring that only the most relevant concepts are selected for inclusion in speech.

All propositional speech relies on the retrieval and use of semantic knowledge. This is true at the lexical level, since the selection of words for production is guided by their meaning. But it is also true at the broader conceptual level, as the content of our speech is informed by our general semantic knowledge about the topic under discussion. For example, describing one's favourite season requires access to stored semantic knowledge about the typical characteristics of each time of year, the events associated with them and so on. Thus, the coherence of an individual's discourse is likely to be critically determined by (a) the quality of the semantic knowledge they have on the topic under discussion and (b) by their ability to retrieve and select the most appropriate information to talk about. It is important to note that these elements of semantic processing are served by distinct neural systems. Current theories hold that representations of semantic knowledge are centred on the anterior temporal cortices and that a separate 'semantic control' system provides top-down regulation of the activation and selection of concepts from this store, based on current situational demands (*Badre and Wagner, 2002*; *Hoffman et al., 2018*; *Jefferies, 2013*; *Jefferies and Lambon Ralph, 2006*; *Ralph et al., 2017*; *Yee and Thompson-Schill, 2016*). These principal components of semantic cognition – the store of representations and the control system – can be independently impaired following brain damage (*Jefferies, 2013*; *Ralph et al., 2017*).

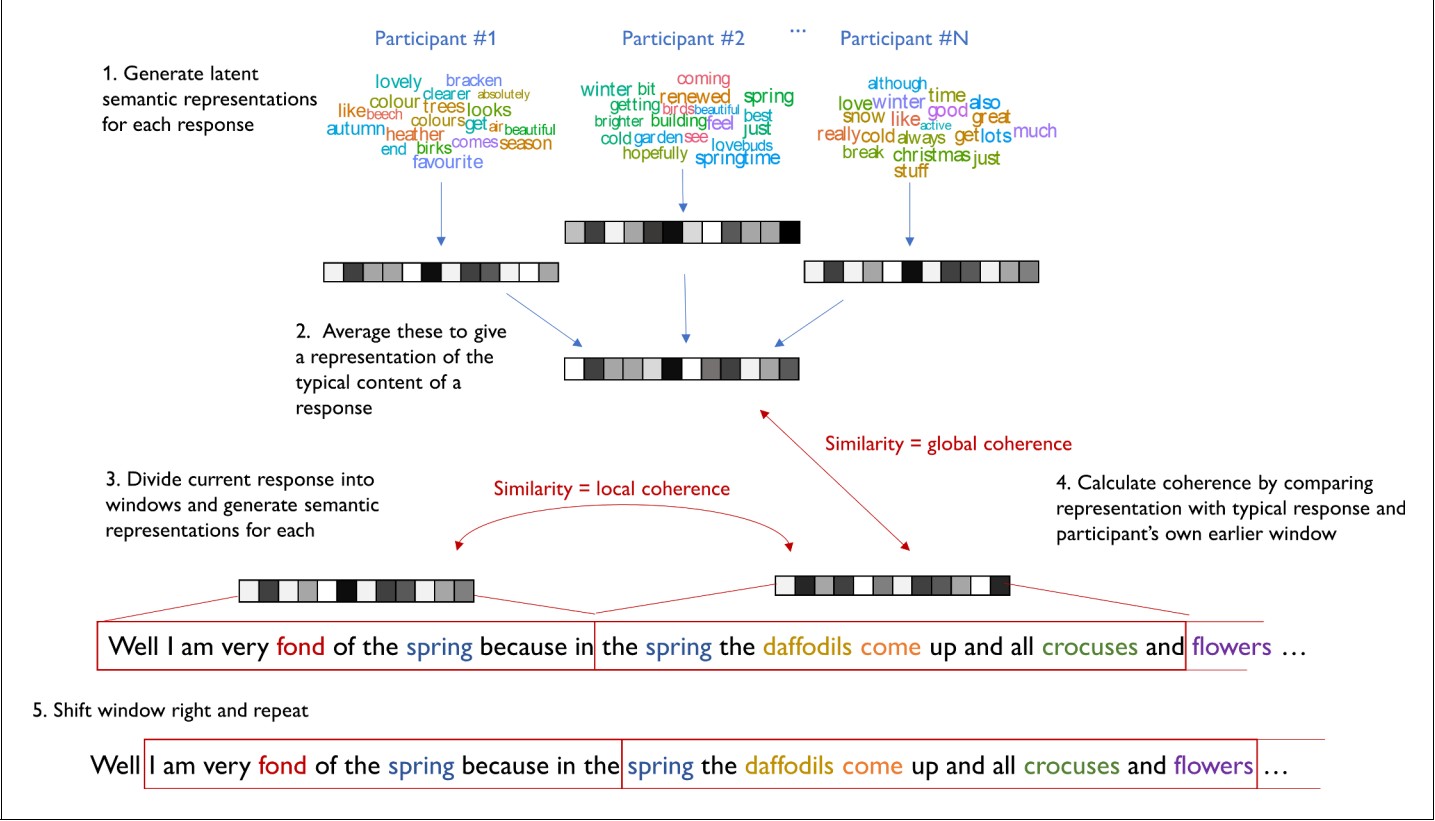

**Figure 1.** Process for computing coherence in speech samples.
DOI: https://doi.org/10.7554/eLife.38907.003

Importantly for the present study, recent evidence suggests some aspects of semantic control are impaired in later life. Hoffman (*Hoffman, 2018*) recently tested the verbal semantic abilities of 100 young and older adults. In common with many previous studies, older people were found to have a broader vocabulary, indicating a richer repository of semantic knowledge (*Grady, 2012*; *Nilsson, 2003*; *Nyberg et al., 1996*; *Park et al., 2002*; *Rönnlund et al., 2005*; *Salthouse, 2004*). Unlike previous studies, we also probed semantic control abilities, using two experimental paradigms commonly used in cognitive neuroscience studies (*Badre et al., 2005*); Whitney, Kirk, *Whitney et al., 2012*). The first task probed the ability to engage in controlled search of the semantic knowledge store to detect weak associations between concepts. No age differences were found for this ability. The second tested the ability to select among competing semantic associations (hereafter termed *semantic selection*). The semantic selection task required participants to inhibit prepotent but irrelevant semantic information in favour of task-relevant aspects of knowledge. Older adults were less successful than young people, indicating age-related decline in this aspect of semantic control. Evidence for age-related decline in controlled selection of semantic information is consistent with a meta-analysis of 47 functional neuroimaging studies indicating that older adults show reduced activity in the left prefrontal region most strongly linked with this ability (*Hoffman and Morcom, 2018*). The ability to select task-relevant semantic representations may be crucial in speech production because it may allow people to select the most relevant aspects of knowledge for use in speech and thus to avoid irrelevant shifts in topic.

Recent studies therefore suggest that ageing is associated with both positive and negative changes in the function of the semantic system. Here, we tested whether these changes could account for age-related declines in the coherence of speech. Young and older adults were asked to produce samples of speech in response to a series of prompts and the coherence of these samples was estimated using a novel computational approach. We hypothesised in particular that individuals with reduced semantic selection abilities would produce less coherent speech, since they would be

less able to prevent irrelevant semantic information from influencing their responses. Importantly, we tested whether semantic abilities had unique effects on coherence, after accounting for the effects of domain-general executive function. We used the Trails test as a measure of domain-general executive function because it is a well-established task which draws on various aspects of executive control including task-switching and inhibition (*Arbuthnott and Frank, 2000*; *Salthouse, 2011*; *Sánchez-Cubillo et al., 2009*) and also because it has previously been linked to poor coherence in speech (*Arbuckle and Gold, 1993*; *Wright et al., 2014*). Finally, we also investigated the production of speech under conditions of divided attention, by including a dual-task condition in which participants completed a secondary manual task while speaking. We included this condition because a previous study has shown that people produce less coherent speech when their attention is divided and that this effect interacts with age (*Kemper et al., 2010*).

## Results

### Assessments of cognitive and semantic ability

Mean scores on a series of background cognitive tests are reported in *Appendix 1—table 1*. Young people were faster to respond in the Trails test and produced more items in category fluency. Older people produced slightly more words in letter fluency, however. There were no group differences in MMSE scores, with all participants scoring at least 26/30.

Participants completed a series of semantic tasks that probed semantic selection ability, breadth of semantic knowledge and controlled retrieval of weak semantic associations. The full analysis of these tasks is reported in Appendix 5, with mean scores in each condition presented in *Appendix 5—figure 1*. The older group scored significantly higher on the vocabulary tests of semantic knowledge, indicating that they had a broader set of verbal semantic information available to them. Controlled retrieval was assessed by manipulating association strength during semantic judgements,

**Table 1.** Results of mixed effects models predicting global and local coherence.

| | Model 1 | | | Model 2 | | | Model 3 | | |
|---|---|---|---|---|---|---|---|---|---|
| | B | Se | P | B | Se | P | B | Se | P |
| *Global Coherence* | | | | | | | | | |
| (Intercept) | 44.6 | 1.45 | <0.001 | 44.6 | 1.43 | <0.001 | 44.6 | 1.42 | <0.001 |
| Age | −2.30 | 0.54 | <0.001 | −1.97 | 0.50 | <0.001 | −0.68 | 0.73 | .35 |
| Task | −0.31 | 0.24 | .20 | −0.31 | 0.24 | .20 | −0.30 | 0.24 | .21 |
| Age*Task | −0.53 | 0.25 | .056 | −0.53 | 0.25 | .052 | −0.53 | 0.25 | .052 |
| Response length | −0.97 | 0.38 | .014 | −0.90 | 0.36 | .016 | −0.86 | 0.35 | .019 |
| Trails ratio | | | | −1.61 | 0.45 | <0.001 | −1.71 | 0.42 | <0.001 |
| Semantic knowledge | | | | | | | −1.63 | 0.73 | .028 |
| Semantic selection | | | | | | | 1.16 | 0.42 | .007 |
| Weak association | | | | | | | 0.29 | 0.52 | .58 |
| *Local Coherence* | | | | | | | | | |
| (Intercept) | 26.8 | 1.36 | <0.001 | 26.8 | 1.35 | <0.001 | 26.8 | 1.34 | <0.001 |
| Age | −2.51 | 0.58 | <0.001 | −2.27 | 0.57 | <0.001 | −1.42 | 0.80 | .081 |
| Task | −0.16 | 0.34 | .63 | −0.16 | 0.33 | .63 | −0.16 | 0.34 | .64 |
| Age*Task | −0.55 | 0.32 | .10 | −0.55 | 0.32 | .10 | −0.55 | 0.32 | .10 |
| Response length | 0.01 | 0.39 | .98 | 0.09 | 0.38 | .82 | 0.12 | 0.38 | .74 |
| Trails ratio | | | | −1.16 | 0.45 | .012 | −1.23 | 0.43 | .006 |
| Semantic knowledge | | | | | | | −1.14 | 0.76 | .14 |
| Semantic selection | | | | | | | 0.94 | 0.43 | .034 |
| Weak association | | | | | | | 0.33 | 0.54 | .54 |

DOI: https://doi.org/10.7554/eLife.38907.004

since the detection of weak associations requires greater control over the retrieval of information from semantic memory (*Badre and Wagner, 2007*). This manipulation had similar effects in young and older people, suggesting that the ability to retrieve less salient semantic knowledge was equivalent in the two groups. Semantic selection was probed using a task in which participants were asked to match items based on particular semantic features (e.g., colour) (*Thompson-Schill et al., 1997*). Selection demands were highest when the correct target was incongruent with pre-existing semantic associations (e.g., *salt* goes with *snow*, not *pepper*). Older people showed a larger effect of the congruency manipulation, performing more poorly in the incongruent condition. This indicates that the older group had difficulty in selecting task-relevant semantic knowledge and inhibiting irrelevant associations.

## Speech rate

We now turn to analyses of the speech samples produced by participants. We first considered the effect of our experimental manipulations on rate of speech production (number of words produced per minute). The results are shown in *Figure 1A*. Mixed effects modelling indicated that speech rate was influenced both by age group ($B = -7.74$, *se* = 3.81, p=0.046), with older participants tending to speak more slowly, and by task ($B = -1.93$, *se* = 0.79, p=0.016), with fewer words produced under dual-task conditions. We therefore included speech rate as a covariate in subsequent analyses, to ensure that effects on coherence were not attributable to this variable.

## Predictors of coherence

Coherence of speech was assessed using a novel computational approach (see Materials and methods and *Figure 1*). Measures of global coherence (GC) and local coherence (LC) were computed. We began by assessing the internal reliability of the GC and LC measures over the fourteen prompts used to elicit speech samples. Cronbach's alpha was high for both measures (GC = 0.83; LC = 0.79), indicating that stable individual differences in coherence were present over the various topics about which participants were asked to speak. GC and LC values were also strongly correlated with one another (r = 0.79), suggesting that both are closely linked, as found in previous studies. In the older group, age was negatively correlated with GC (r = −0.64) and LC (r = −0.56).

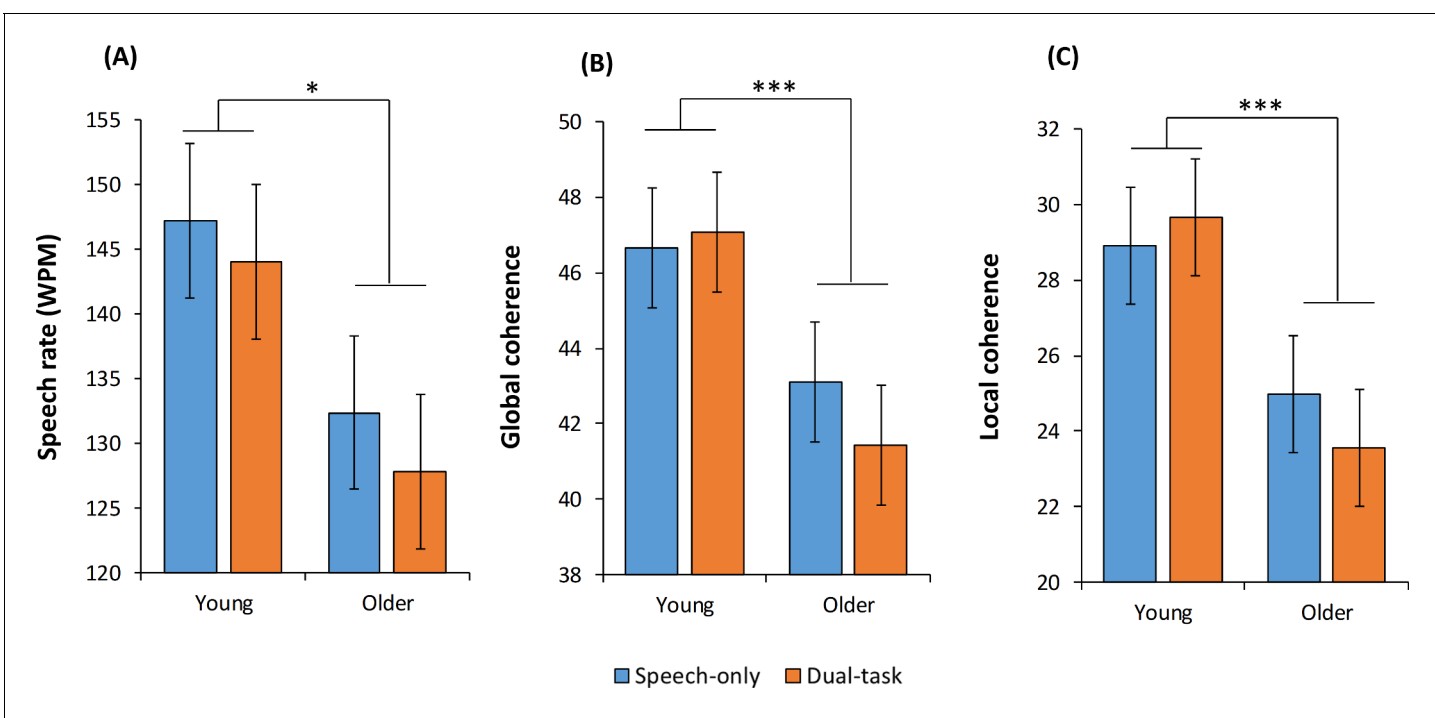

**Figure 2.** Effects of age and task on speech rate and coherence *p<0.05; ***p<0.001.
DOI: https://doi.org/10.7554/eLife.38907.005

Next we investigated the effects of our experimental manipulations on the GC of participants' speech. The first model included age group and task as predictors, with speech rate as an additional covariate (see *Table 1* for results). Age group was a strong predictor of GC: as predicted, older participants produced markedly less coherent speech than young people (see *Figure 2B*). As shown in *Table 1*, the dual-task manipulation had no effect on GC. The interaction between age and task fell just short of statistical significance (p=0.052). This suggests that there may be a weak tendency for the effect of the task manipulation to be larger in older people. Speech rate was a negative predictor of GC, indicating that participants who spoke more quickly showed a greater tendency to deviate from the topic being probed.

The addition of Trails ratio scores (Model 2) significantly improved the fit of the model ($\chi^2(1) = 11.9$, p<0.001). As expected, participants with a smaller ratio of Trails B to A (indicating better executive ability) had higher GC values. However, this was not sufficient to explain the lower GC values of older people: a significant difference between the young and older groups remained. The inclusion of semantic test scores (Model 3) yielded a further improvement in model fit ($\chi^2(3) = 10.1$, p=0.018). The estimated effects of the test scores on GC are plotted in *Figure 3*. Participants with higher scores on the semantic selection test produced more coherent speech. The effect of Trails ratio was also significant and there was also a tendency for individuals with higher semantic knowledge scores to produce *less* coherent speech. Weak association task scores were not a significant predictor of GC. Importantly, there was no remaining effect of age group in this model (see *Table 1*), suggesting that lower levels of GC in older adults can be explained in terms of their lower semantic selection and higher semantic knowledge scores.

LC measurements were subjected to the same sequence of analyses, with broadly similar results (see *Table 1*). The first model revealed an effect of age group with no effect of task and a non-significant interaction (see *Figure 2C*). In Model 2, Trails ratio was again a significant predictor of coherence, but a significant age effect remained. In contrast, the age group effect was not significant once semantic scores were included (Model 3). Scores on the semantic selection task were a significant predictor of LC, with participants who performed poorly on this task tending to be less coherent (see *Figure 3*).

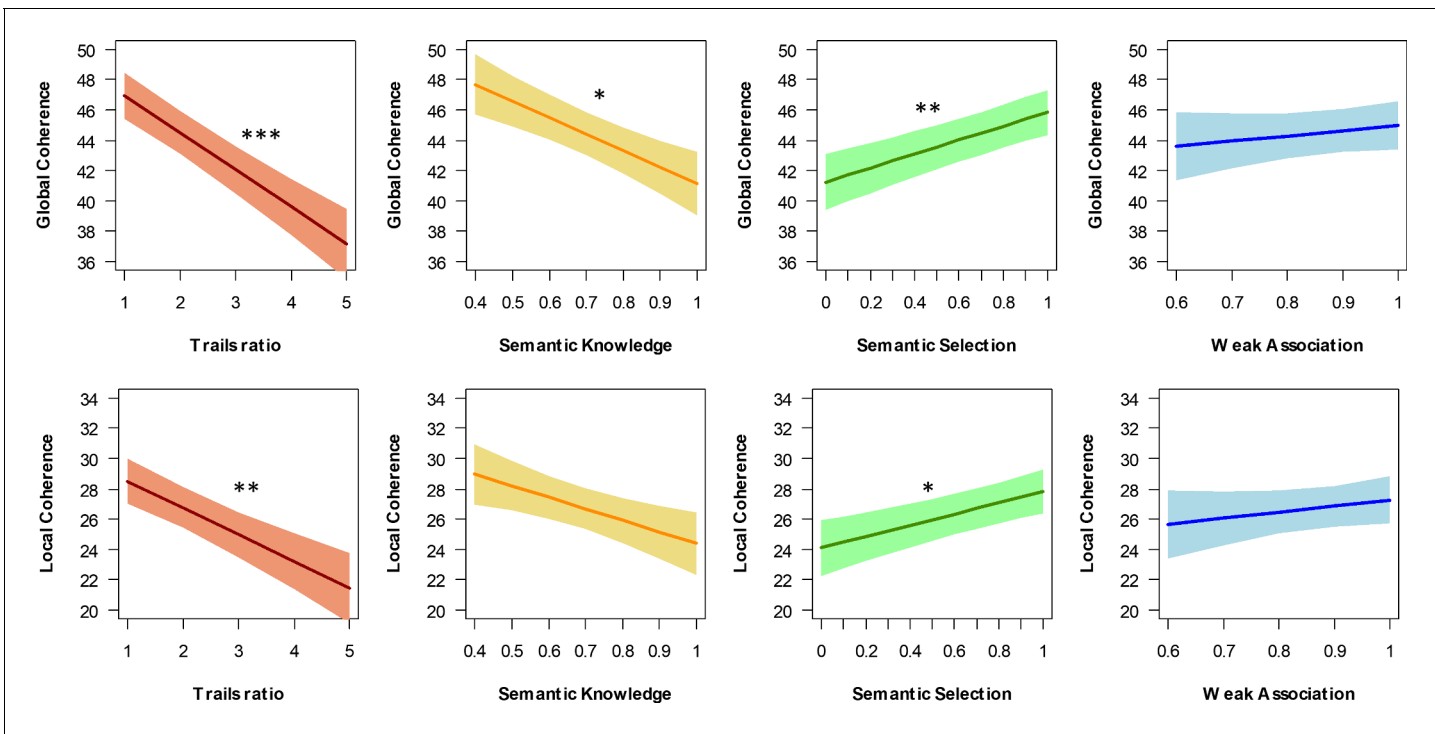

**Figure 3.** Effects of semantic and executive abilities on coherence *p<0.05; **p<0.01; ***p<0.001.
DOI: https://doi.org/10.7554/eLife.38907.006

## Predictors of other speech measures

The purpose of this analysis was to establish whether the observed effects of executive and semantic abilities on coherence were specific to this aspect of speech, or whether they would be observed for other characteristics of speech. Principal components analysis was used to reduce the nine properties of speech into four latent factors, shown in *Figure 4A*. These were the only factors with eigenvalues greater than one and together they explained 81% of the variance. GC and LC loaded exclusively on Factor 2, confirming that coherence emerged as a discrete characteristic of speech. Factor 1 indexed the use of long, abstract, late-acquired words in speech, so appeared to reflect access to complex vocabulary. High scores on Factor 3 were associated with use of low frequency, concrete nouns that were low in semantic diversity. This factor may reflect the degree to which speech referenced highly specific concepts, so we labelled it semantic specificity. Finally, high scores on Factor 4 were characterised by high type:token ratio and a low proportion of closed-class words, which are indicative of greater lexical diversity.

Scores on each factor were subjected to the same series of mixed effects analyses used for the analysis of GC and LC. The full results of these analyses are shown in *Supplementary file 1*, while the effects of participants' semantic and executive scores on each factor are presented in *Figure 4B*. The results for Factor 2 (coherence) were the same as previously observed for GC and LC separately: lower coherence was associated with poorer Trails and semantic selection performance but with better semantic knowledge. Importantly, no other factor showed the same pattern. The only other

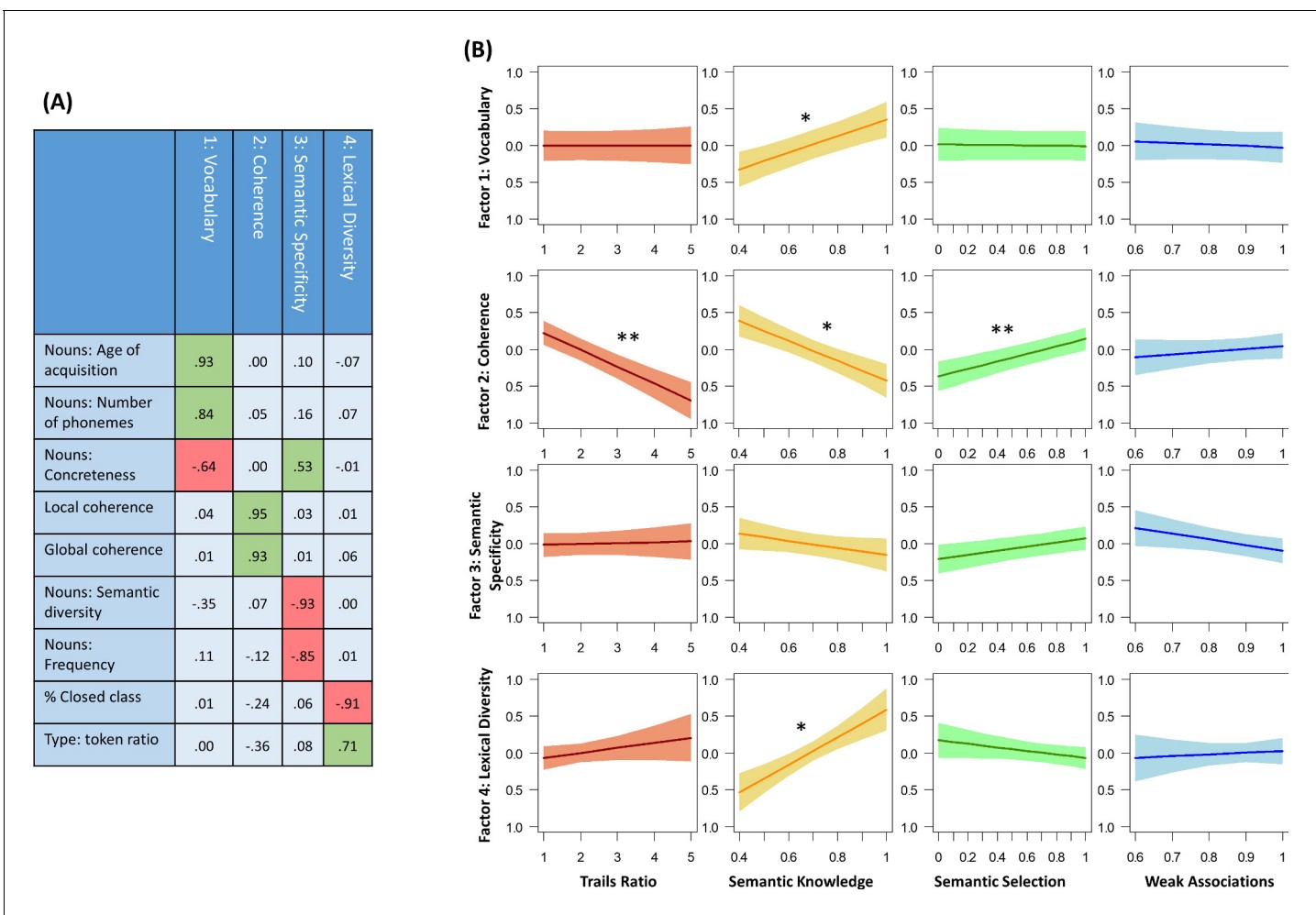

**Figure 4.** (A) Principal components analysis identifying four latent speech factors. (B) Effects of semantic and executive abilities on factor scores. *p<0.05; **p<0.01; ***p<0.001.
DOI: https://doi.org/10.7554/eLife.38907.007

significant effects were that semantic knowledge was positively correlated with scores on factors 1 (vocabulary) and 4 (lexical diversity), indicating that participants with broader semantic knowledge used a broader and more complex range of vocabulary when speaking. This analysis therefore confirms that the participants' executive and semantic selection abilities had a specific effect on their coherence but not on other aspects of their speech production.

### Secondary task performance

Analysis of the secondary manual task is reported in Appendix 5. In brief, older people had slower RTs and both groups were slower when the task was combined with speaking. Importantly, however, there was no interaction between these factors, indicating that the requirement to perform two tasks affected both groups equally (see *Appendix 5—figure 2*). Participants' GC and LC scores did not predict performance on the secondary task, ruling out the possibility of a trade-off between secondary task performance and maintenance of coherence.

## Discussion

The ability to produce coherent speech is critical for effective communication but tends to decline in later life. Here, we investigated cognitive factors that predict this decline, using computational linguistic techniques to quantify the coherence of speech produced by a large group of young and older adults. We replicated previous findings indicating that individuals with greater domain-general executive ability produce more coherent speech, but this effect did not fully account for age differences in coherence. However, when we included semantic abilities as additional predictors of coherence, the age group difference was eliminated. Semantic selection ability emerged as a positive predictor of coherence while breadth of semantic knowledge was a negative predictor. These effects were specific to coherence and not to other characteristics of speech, and were not attributable to differences in speech rate. Our results indicate that older people produce less coherent speech (a) because they are less skilled at selecting the most relevant aspects of semantic knowledge to include in their speech and (b) because they have a larger set of semantic knowledge to select from.

First and foremost, our results establish that the monitoring and control of discourse is influenced by the function of the semantic system, in addition to domain-general executive resources. In particular, we found that the ability to select task-relevant semantic information was a strong predictor of coherence in speech. The task we used to assess this ability is well-established as a measure of semantic control (e.g., *Badre et al., 2005*; *Thompson-Schill et al., 1997*; *Whitney et al., 2012*) and required participants to attend to specific semantic features of objects while inhibiting strong but irrelevant semantic associations. Our data indicate that similar selection and inhibitory demands are present during the production of discourse. A conversational cue, such as 'what's your favourite season?', initially causes a wide range of knowledge to become activated in the semantic system. Some of this information will be useful in answering the question and some less so. Coherent communication requires the speaker to select the subset of that information which is directly relevant at the current time, while suppressing aspects of knowledge that are activated but less pertinent. These demands grow as the narrative develops and new associations are activated.

Of course, the knowledge that drives speech production is not solely semantic in nature – specific episodic memories and more general autobiographical knowledge will often be triggered as well. The prompts used in the present study were designed to elicit general knowledge rather than specific personal experiences. However, episodic and semantic memories are mutually interdependent (*Binder et al., 2009*), and it was clear that participants drew on both in generating their responses. It is likely that selection mechanisms for these distinct types of memory are shared to some degree. Indeed, the left mid ventrolateral prefrontal cortex (VLPFC), the brain region most closely associated with semantic selection, responds to selection demands in all three domains (*Badre and Wagner, 2007*; *Dobbins and Wagner, 2005*; *King et al., 2005*). In particular, a large literature has examined brain regions implicated in the selection of task-relevant aspects of retrieved episodic memories. These processes are often referred to as 'post-retrieval monitoring'. The requirement to selectively recall particular details of an event drives greater activation in left mid VLPFC, suggesting that this area also mediates selection from episodic memory (*Badre and Wagner, 2007*; *Dobbins and Wagner, 2005*). However, the monitoring and selection of retrieved episodic memories is also associated with activation of dorsolateral prefrontal regions that are not implicated in selection from semantic

knowledge (*Fletcher et al., 1998*; *Rugg et al., 2003*), which may indicate a degree of independence between semantic and episodic selection at the neural level.

At a behavioural level, recent work indicates that patients with deficits in semantic selection also find it hard to resolve interference in episodic memory tasks (*Stampacchia et al., 2018*). In addition, healthy older people often show increased interference from irrelevant events when retrieving episodic memories (*Campbell et al., 2010*; *Ikier et al., 2008*). Combined with our previous study (*Hoffman, 2018*), these findings point to a more general old-age deficit in selecting the most task-relevant aspects of retrieved semantic *and* episodic knowledge. We have demonstrated here that this selection deficit contributes to the loss of coherence in later life. This conclusion is consistent with age-related changes in the structure and function of the VLPFC region most associated with this ability. Lateral prefrontal cortex exhibits the greatest reductions in cortical volume as a function of age (*Fjell et al., 2009*; *Raz et al., 2004*). In addition, a recent meta-analysis of 47 functional neuroimaging studies found that older adults activated this region less strongly than young people during semantic processing (*Hoffman and Morcom, 2018*).

Another important question is the extent to which the selection processes involved in regulating semantic knowledge overlap or interact with other, domain-general executive functions. This is an area of active debate, with researchers proposing that some aspects of the regulation of semantic knowledge are performed by domain-general systems for competition resolution while others require more specialised neural resources (*Badre et al., 2005*; *Jefferies, 2013*; *Nagel et al., 2008*; *Whitney et al., 2012*). In this study, we employed a single measure of non-semantic executive ability, the Trails test. The measure of executive ability derived from this test was not correlated with performance on the semantic selection task (young group: $r = 0.08$; older group: $r = 0.02$) and, although Trails performance was a strong predictor of coherence, we found that semantic selection had an additional, independent effect. This suggests that the relationship between coherence and semantic selection cannot simply be attributed to poorer general executive ability.

However, it is unlikely that a single test can adequately capture all aspects of executive function. There are many different views on how executive functions are organised, but one common scheme proposes separate shifting, updating and inhibition components (*Miyake et al., 2000*). The Trails test primarily taps shifting (or task-switching) ability (*Arbuthnott and Frank, 2000*; *Hedden and Yoon, 2006*; *Sánchez-Cubillo et al., 2009*). In future, it will be important to probe potential contributions of other components of executive function, in particular inhibition. It is currently unclear whether suppression of irrelevant semantic information retrieved from memory involves the same executive resources as inhibition of overt behavioural responses, as measured by paradigms such as the Go/No Go task (*Verbruggen and Logan, 2008*). This is an important issue to resolve if we are to understand how semantic selection processes relate to domain-general executive function, particularly as semantic selection deficits in later life may be related to more general declines in inhibitory function, which have been reported across a range of tasks (*Borella et al., 2008*; *Hasher and Zacks, 1988*; *Hoffman, 2018*; *Salthouse and Meinz, 1995*).

We also found that the breadth of participants' semantic knowledge influenced their coherence. Individuals with a wider range of lexical-semantic knowledge tended to be *less* coherent. This effect is consistent with the notion that selecting appropriately from activated knowledge is critical to maintaining coherence. This challenge becomes greater the more information one has in one's semantic store, simply because more concepts are likely to be activated in response to any given cue. Thus, our data indicate that being more knowledgeable in itself brings greater challenges in identifying the most relevant aspects of knowledge to use in speech. It is very well-established that older people have greater semantic and general world knowledge, as was the case in our study (*Rönnlund et al., 2005*; *Salthouse, 2004*; *Verhaeghen, 2003*), so this factor may also have contributed to age-related coherence declines. It is also worth noting that breadth of semantic knowledge also predicted the use of more sophisticated vocabulary (i.e., more late-acquired, abstract nouns) and greater lexical diversity (for a similar result, see *Kemper and Sumner, 2001*). Therefore, it appears that quantity of semantic knowledge has more global effects on the characteristics of speech, unlike semantic selection, which impacts specifically on coherence.

We found that conditions of divided attention had no overall effect on the coherence of speech, in contrast to the findings of *Kemper et al., 2010*. However, we note that our secondary task appeared less demanding that that used by Kemper and colleagues. Our secondary task produced a reduction in speech rate of around 5 WPM, compared with 20–40 WPM in Kemper et al. Despite

this, we did find a non-significant trend (p=0.056) towards an interaction of dual-task demands with age (for the GC measure). There was a suggestion that the secondary task may have had a small effect on the coherence of older people, while young people appeared unaffected (see *Figure 2B*). There remains a possibility therefore that divided attention has a particular detrimental effect on coherence in older adults. This could have important implications for conversations conducted in everyday situations in which speakers may simultaneously be engaged in other activities (e.g., talking while driving, shopping etc.). Future studies with more demanding concurrent tasks are needed to assess this possibility and its interaction with semantic abilities.

Previous studies have found varying effects of age on coherence, depending on the speech elicitation task used. Typically, narratives elicited from verbal prompts, as in the present study, reveal the greatest decrements in coherence while tasks that elicit speech using visual stimuli, such as picture descriptions or story-telling from comic strips, produce smaller effects (*James et al., 1998*; *Wright et al., 2014*). These results fit well with our assertion that the ability to select relevant semantic content is a major determinant of coherence. When a pictured stimulus is used to cue speech, it acts as a source of constraint over semantic activation. Upon analysing the image, knowledge related to the objects and events depicted automatically comes to mind and can be used to drive speech production. If any irrelevant concepts become activated during this process, they can easily be eliminated on the basis that they are not present in the image. In contrast, constructing a response to a brief verbal prompt is a trickier proposition, since a wide range of potentially relevant information may be activated and no external cues are available to guide selection. Of course, the monologues elicited in the present study are a rather extreme example of this phenomenon. In everyday conversational speech, environmental cues are often available to guide the selection of speech content. For instance, a look of confusion from the speaker's interlocutor can indicate when a loss of topic has occurred and a well-timed question could direct the speaker back to the topic in hand. Such cues can only be effective if speakers are sensitive to them, however, and evidence suggests that speakers with poor coherence are also less skilled at interpreting social cues (*Pushkar et al., 2000*).

Finally, it is important to consider an underlying assumption often made in the literature on coherence: that greater coherence is always a desirable characteristic for speech. Many situations do require specific information to be communicated quickly and efficiently and in these cases, it is beneficial to be able to provide the most germane information without digression to other topics. For example, there is evidence that individuals who are less coherent in conversation perform poorly at communicating task-related information to a partner in an experimental setting (*Arbuckle et al., 2000*). In other situations, however, a less focused approach to speech may have its advantages. When the goal of a conversation is to entertain, rather than to convey specific information, the ability to shift focus away from the original topic may be beneficial. Indeed, older people are generally considered to produce more enjoyable stories than young people (*Ryan et al., 1992*). One notable study collected responses of young and older people to questions about life events and asked judges to rate them on various dimensions (*James et al., 1998*). Older people produced more off-topic speech than young people and their narratives were rated as less focused. However, while less coherent speakers were rated as less clear and focused, they were also considered to be more interesting and to have produced better stories. In summary, the ability to communicate coherently is critical in many but not all everyday conversations. It is possible that the most effective communicators are those who can tailor their selection of content to the current situational demands, focusing tightly on the subject at hand when required to but broadening their focus at other times. Little is known at present about how coherence interacts with these situational demands and this is one area where more research is needed.

## Materials and methods

### Participants

Thirty young adults, aged between 18 and 30 (mean = 19.3), were recruited from the undergraduate Psychology course at the University of Edinburgh and participated in the study in exchange for course credit. Thirty older adults, aged between 61 and 91 (mean = 76.0), were recruited from the Psychology department's volunteer panel. These participants were a subset of a group taking part in a larger study of semantic processing, some data from which have been reported elsewhere

(*Hoffman, 2018*). All participants reported to be in good health with no history of neurological or psychiatric illness. Demographic information for each group is shown in *Appendix 1—table 1*. Young and older adults did not differ significantly in years of education completed ($t$(58) = 0.93, p=0.36). Sample size was selected to be similar to comparable studies in the literature. All participants provided informed consent and the study was approved by the University of Edinburgh Psychology Research Ethics Committee (120-1415/3).

## Assessments of cognitive and semantic ability

Participants completed the following tests of general cognitive function and executive ability: Mini-Mental State Examination, Trail-making task, verbal fluency (see Appendix 1 for details). As a measure of domain-general executive function, we computed the ratio of completion times for Trails part B to Trails part A. High ratios indicated disproportionately slow performance on part B, indicative of poor executive function. A ratio rather than a difference score was used as this measure minimises the influence of differences in general processing speed (*Arbuthnott and Frank, 2000*; *Salthouse, 2011*). Participants also completed two tasks probed breadth of semantic knowledge: lexical decision (*Baddeley et al., 1992*) and synonym matching (adapted Mill Hill vocabulary scale; *Raven et al., 1989*) (for further details, see Appendix 1). As scores on these tasks were strongly correlated, they were averaged to give a single measure of breadth of semantic knowledge.

Semantic control was assessed using a 2 × 2 within-subjects experimental design that manipulated the need for semantic control in two different tasks (*Hoffman, 2018*); following *Badre et al., 2005*). In the first task, participants made semantic decisions based on global semantic association. They were presented with a probe word and asked to select its semantic associate from either two or four alternatives (see *Figure 5* for examples). The strength of association between the probe and target was manipulated: the associate was either strongly associated with the probe (e.g., *town-city*) or more weakly associated (e.g., *iron-ring*). The Weak Association condition was assumed to place greater demands on controlled retrieval of semantic information, because automatic spreading of activation in the semantic network would not be sufficient to identify the correct response (*Badre and Wagner, 2007*). In the second task, participants were asked to select items that matched on particular features. At the beginning of each block, participants were given a feature to attend to (e.g., Colour). On each trial, they were provided with a probe and were asked to select the item that was most similar on the specified feature. Trials manipulated the semantic congruency of the probe and target. On Congruent trials, the probe and target shared a pre-existing semantic relationship, in addition to matching on the currently relevant feature (e.g., *cloud-snow*). In contrast, on Incongruent trials the probe and target shared no meaningful relationship, other than matching on the specified feature (e.g., *salt-dove*). Furthermore, on these trials one of the foils had a strong semantic relationship with the probe, although it did not match on the currently relevant feature (*salt-pepper*). Incongruent trials placed high demands on semantic selection processes for two reasons: first, because there was no pre-existing semantic relationship between probe and target to boost activation of the target and second, because the strong but irrelevant relationship between the probe and foil had to be ignored.

## Speech elicitation task

Samples of speech were elicited under conditions of undivided and divided attention. On *speech-only* trials, participants were asked to speak for 60 s at a time in response to a written prompt (for full list of prompts, see Appendix 2). Prompts were designed to probe particular areas of semantic knowledge (e.g., *What sort of things do you have to do to look after a dog?*). Participants read each prompt on a computer monitor and pressed a key when ready to begin speaking. After 60 s, a tone sounded to signal the end of the trial. Participants were instructed to continue speaking until they heard the tone. On *dual-task* trials, participants were asked to complete an attention-demanding secondary task while speaking (*Craik et al., 1996*). On these trials, a horizontal array of four squares appeared on screen. Every 3 s, a red circle appeared in one of the squares and participants pressed a key corresponding to its location. This task was performed continuously throughout the speech elicitation period. Seven speech samples were obtained in the speech-only condition and seven in the dual-task condition. Finally, to obtain a baseline measure of secondary task performance, there

# Breadth of semantic knowledge

### Which is the real word?

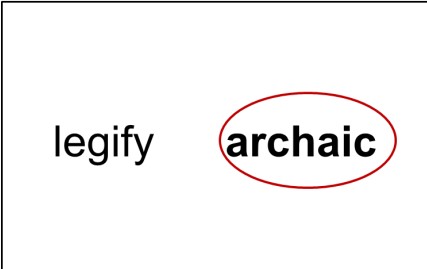

### Which means the same as bombastic?

bombastic

destructive **pompous**
anxious  bickering

# Controlled retrieval ability

### STRONG

### Which is related to town?

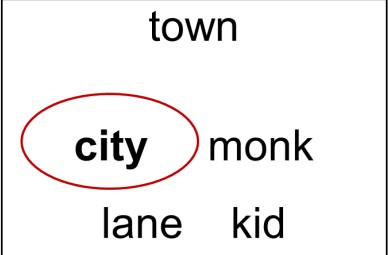

### WEAK

### Which is related to iron?

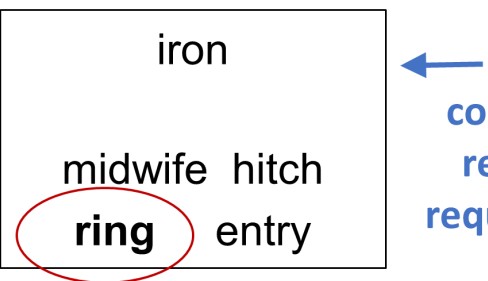

← **High controlled retrieval requirement**

# Semantic selection ability

### CONGRUENT

### Which is the same colour as cloud?

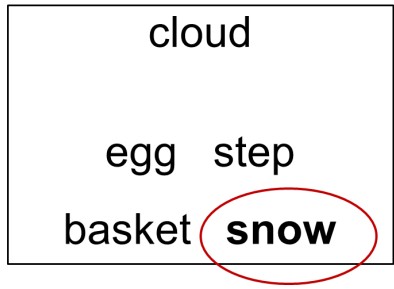

### INCONGRUENT

### Which is the same colour as salt?

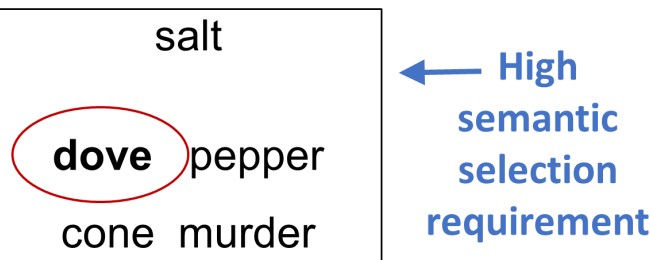

← **High semantic selection requirement**

**Figure 5.** Example trials from semantic tasks. The correct response is highlighted in each case.
DOI: https://doi.org/10.7554/eLife.38907.008

were five *secondary-only* trials where the secondary task was performed without speech for 60 s. These trials were interspersed amongst the speech elicitation trials.

Spoken responses were digitally recorded for later transcription. The main dependent variables analysed were computed measures of global and local coherence (GC and LC), as described below. A number of other speech markers were also computed and were included in supplementary analyses (see Appendix 3 for details).

## Coherence computations

Measures of local and global coherence were generated using an automated computational linguistic approach. Analyses were implemented in R; the code is publicly available and can easily be applied to new samples (https://osf.io/8atfn/). Our approach used latent semantic analysis (LSA) (*Landauer and Dumais, 1997*), one of a number of computational techniques in which patterns of word co-occurrence are used to construct a high-dimensional semantic space. The LSA method utilises a large corpus of natural language divided into a number of discrete contexts. The corpus is used to generate a co-occurrence matrix registering how often each word appears in each context. Data reduction techniques are then applied to this matrix, with the result that each word is represented as a high-dimensional vector. Words that are used in similar contexts (and are thus assumed to have related meanings) are assigned similar vectors. Word similarities derived in this way are strong predictors of human judgements of semantic relatedness and human performance on a range of tasks (*Bullinaria and Levy, 2007*; *Recchia and Jones, 2009*).

Importantly, vectors for individual words can be combined linearly to represent the meanings of whole sentences and passages of speech/text (*Foltz et al., 1998*). A number of researchers have used this property to generate estimates of coherence for texts or spoken samples based on LSA similarity measures (*Elvevåg et al., 2007*; *Foltz, 2007*; *Foltz et al., 1998*; *Graesser et al., 2004*). The present work builds on this approach. The overall strategy we took was to divide each speech sample into smaller windows (of 20 words each) and to use LSA to generate vector representations of the semantic content of each window. Coherence was assessed by measuring the similarity of the vector for each window with that of the previous window (LC) and with a vector representing the typical semantics of responses to the same prompt (GC). This process is illustrated in *Figure 1*.

First, an LSA representation of each participant's response was computed by averaging the LSA vectors of all the words they produced in response to the same prompt (for details of the averaging method and vector space used, see Appendix 3). These were averaged to give a composite vector that represented the typical semantic content produced in response to that prompt (this step excluded the target response). Next, the target response was analysed using a moving window approach. The target response was divided into windows of 20 words in length. An LSA vector was computed for each window. Local coherence was defined as the cosine similarity of the semantic vector for the current window with that of the previous window. Therefore, in common with other researchers (*Elvevåg et al., 2007*; *Foltz, 2007*), we define LC as the degree to which adjoining utterances convey semantically related content. A low LC value would be obtained if a participant switched abruptly between topics during their response.

Global coherence was defined as similarity of the vector for each window with the composite vector derived from the other participants' responses. Therefore, GC was a measure of how much the target response matched the typical semantic content of responses to that prompt. A low GC value would be obtained if a participant tended to talk about other topics that were semantically unrelated to the topic being probed. Thus, our measure of GC captured the degree to which participants maintained their focus on the topic under discussion, in line with the definition used by other researchers (*Glosser and Deser, 1992*; *Wright et al., 2014*).

Once GC and LC had been calculated, the window moved one word to the right and the process was repeated, until all windows had been assessed. GC and LC values were averaged across windows to give overall values for each response, which were multiplied by 100 for ease of presentation. Examples of responses with high vs. low coherence values are provided in *Appendix 3-Table 1*. The LSA-based coherence measures were validated by comparing them with judgements of coherence provided by human raters for a subset of speech samples (see Appendix 4 for details). There was a strong correlation between rated GC and LSA-based GC ($r = 0.68$) and a somewhat weaker relationship between LSA-based LC and LC ratings ($r = 0.37$). Test-retest reliability was high (see Appendix 4).

## Statistical analyses

A series of linear mixed effects models were used to investigate the effects of the experimental manipulations and individual differences in semantic and executive ability on characteristics of speech. The dependent variable in the first analysis was speech rate in words per minute (WPM). This was analysed in a linear mixed model with a 2 × 2 (age group x task) factorial design. We performed this analysis because previous studies have found that older people speak more slowly than young people and that speech rate is reduced under dual-task conditions (*Kemper et al., 2003*; *Kemper et al., 2010*). It was important to investigate this possibility in our data as speech rate might have an impact on coherence. For example, participants who spoke very quickly could cover a wider range of topics in 60 s, increasing the likelihood that their response would lose coherence. Since we found that speech rate was indeed influenced by both age and task, this variable was included as a covariate in later analyses.

Our main hypotheses were tested with a series of nested models which used GC and LC as dependent variables (in parallel). The first model included age group and task as predictors, as well as speech rate. Next, we added the Trails ratio score as an additional predictor, to test the hypothesis that general executive ability influences coherence. In the final model, we added three semantic task scores as predictors, to test the hypothesis that semantic abilities are an additional important determiner of coherence. The semantic task scores included were:

1. Semantic knowledge: the mean of accuracy on the lexical decision and synonym matching tasks.
2. Semantic selection: accuracy on the Incongruent condition of the feature association task, which required participants to select task-relevant aspects of semantic knowledge and inhibit irrelevant associations.
3. Weak association: accuracy on the Weak Association condition of the global association task, which required controlled retrieval of semantic knowledge to identify less salient semantic relationships.

Next, to establish whether the observed effects were specific to coherence, we investigated whether other characteristics of speech showed similar effects. We computed six measures of the lexical characteristics of the words produced in each speech sample (see *Figure 4* and Appendix 3). Principal components analysis was performed on these (along with the coherence measures) and used to extract four underlying factors, which were promax-rotated. Scores on each of these factors were then analysed using the same series of nested mixed models employed in the main analysis of GC and LC.

Finally, to analyse performance on the concurrent secondary task, we used a linear mixed model with group and task (secondary-only vs. dual-task) as predictors. The dependent variable was RT. We then added GC and LC values to the model as predictors to determine whether coherence was related to secondary task performance. All study data are available online (https://osf.io/8atfn/).

Mixed effects models were constructed and tested using the recommendations of *Barr et al. (2013)*. We specified a maximal random effects structure, including random intercepts for participants and prompts as well as random slopes across participants for the effect of task and random slopes across prompts for task and age group. Continuous predictors were standardised prior to entry in the model. The significance of fixed effects was assessed by comparing the full model with a reduced model that was identical in every respect except for the exclusion of the effect of interest. Likelihood-ratio tests were used to determine whether the inclusion of the effect of interest significantly improved the fit of the model.

## Acknowledgements

PH was supported by The University of Edinburgh Centre for Cognitive Ageing and Cognitive Epidemiology, part of the cross council Lifelong Health and Wellbeing Initiative (MR/K026992/1). Funding from the Biotechnology and Biological Sciences Research Council (BBSRC) and Medical Research Council (MRC) is gratefully acknowledged. We are grateful to Wing Yee Ho, Eszter Kalapos, Jasmine Kulay, Khushboo Mehra and Adam Ryde for assistance with data collection.

## Additional information

### Funding

| Funder | Grant reference number | Author |
|---|---|---|
| Medical Research Council | MR/K026992/1 | Paul Hoffman |
| Biotechnology and Biological Sciences Research Council | MR/K026992/1 | Paul Hoffman |

The funders had no role in study design, data collection and interpretation, or the decision to submit the work for publication.

### Author contributions

Paul Hoffman, Conceptualization, Software, Formal analysis, Validation, Investigation, Methodology, Writing—original draft, Project administration, Writing—review and editing; Ekaterina Loginova, Validation, Investigation, Writing—review and editing; Asatta Russell, Investigation, Writing—review and editing

### Author ORCIDs

Paul Hoffman (iD) http://orcid.org/0000-0002-3248-3225

### Ethics

Human subjects: All participants provided informed consent and the study was approved by the University of Edinburgh Psychology Research Ethics Committee.(120-1415/3).

### Decision letter and Author response

Decision letter https://doi.org/10.7554/eLife.38907.023
Author response https://doi.org/10.7554/eLife.38907.024

## Additional files

### Supplementary files

• Supplementary file 1. Results of mixed effects models predicting characteristics of speech.
DOI: https://doi.org/10.7554/eLife.38907.009
• Transparent reporting form
DOI: https://doi.org/10.7554/eLife.38907.010

### Data availability

All raw data and code required to replicate the analyses are available at https://osf.io/8atfn/

The following dataset was generated:

| Author(s) | Year | Dataset title | Dataset URL | Database, license, and accessibility information |
|---|---|---|---|---|
| Paul Hoffman | 2018 | Relationship of semantic and executive abilities with coherence in speech | https://osf.io/8atfn/ | Publicly available at OSF (https://osf.io/). |

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

## Appendix 1

DOI: https://doi.org/10.7554/eLife.38907.011

# Details of cognitive testing

## General cognitive assessments

The Mini-Mental State Examination was used as a general cognitive screen. All participants scored 26/30 or above. General executive ability was assessed using the Trail-making task (*Reitan, 1992*). Finally, three categories of verbal fluency were administered, in which participants were given one minute to produce as many words as possible that fit a specific criterion. The criteria included two semantic categories (animals and household objects) and one letter of the alphabet (words beginning with F). Each group's scores on these tasks are presented in *Appendix 1—table 1*.

## Tests of breadth of semantic knowledge

Participants completed two tasks designed to probe the size of their store of semantic representations. These tasks probed knowledge of the meanings and identities of unusual words that were expected to be unknown to some members of the population. They therefore indexed the breadth of semantic knowledge available to each individual. The first task (Lexical Decision) was the Spot-the-Word Test from the Speed and Capacity of Language Processing battery (*Baddeley et al., 1992*). This was a lexical decision test comprising 60 pairs of letter strings. On each trial, participants were presented with a real word and a nonword and were asked to select the real word. Nonwords were phonologically and orthographically plausible, encouraging participants to rely on semantic knowledge for decision-making. The second task (Synonyms) was based on the Mill Hill vocabulary test (*Raven et al., 1989*), a multiple-choice test in which participants are asked to select the synonyms of particular words. There are two parallel forms of the test. To increase the difficulty of the task, the hardest 22 trials from each parallel form were combined to make a new 44-item test. The test was presented in a four-alternative choice format.

## Semantic tasks procedure

Semantic tasks were presented on a PC running Eprime 2.0 software. Participants first completed the Lexical Decision and Synonym tasks and then the semantic control experiment. Each task was preceded by a series of practice trials. Accuracy and reaction times (RTs) were recorded. Participants were instructed to respond as quickly as possible while avoiding mistakes. They were encouraged to guess if unsure of the correct response.

**Appendix 1—table 1.** Demographic information and mean test scores for young and older participants.

|  | Young adults | Older adults |
|---|---|---|
| *N* | 30 | 30 |
| Age | 19.3 (2.2) | 76.0 (8.3) |
| Sex M:F | 8:22 | 15:15 |
| Years of education | 13.8 (0.9) | 14.3 (3.0) |
| MMSE /30 | 28.8 (1.0) | 29.2 (1.1) |
| Category fluency (items per category) | 25.4 (5.4)** | 21.1 (5.2) |
| Letter fluency (items per category) | 12.9 (5.0) | 16.4 (7.1)* |

*Appendix 1—table 1 continued on next page*

*Appendix 1—table 1 continued*

|  | Young adults | Older adults |
|---|---|---|
| Trails A errors | 0.1 (0.3) | 0.1 (0.3) |
| Trails B errors | 0.5 (1.0) | 0.9 (1.4) |
| Trails A time (s) | 26.9 (9.1)** | 35.9 (11.7) |
| Trails B time (s) | 46.2 (12.8)*** | 76.3 (34.9) |
| Trails ratio score (B time / A time) | 1.84 (0.69) | 2.12 (0.70) |

Standard deviations are shown in parentheses. Asterisks indicate the significance of t-tests comparing young and older adults. *p<0.05; **p<0.01; ***p<0.001.

DOI: https://doi.org/10.7554/eLife.38907.012

## Appendix 2

DOI: https://doi.org/10.7554/eLife.38907.013

### Prompts used in speech elicitation task

1. What would it have been like to live in the Middle Ages?
2. What would it be like to live in Antarctica?
3. What happens when a storm is forecast in the UK?
4. What do the police do when a crime has been committed?
5. Which is your favourite season and why?
6. What do you like or dislike about Christmas?
7. Do you think it's a good idea to send people to live on Mars?
8. Why do people come to Scotland on holiday?
9. What sort of things do you have to do to look after a dog?
10. Describe a typical visit to a restaurant.
11. What sort of things does the Queen do on a typical day?
12. What are the advantages and disadvantages of going to university?
13. What do people usually do when getting ready for work in the morning?
14. Describe the steps you would need to take if going somewhere by train.

## Appendix 3

DOI: https://doi.org/10.7554/eLife.38907.014

# Processing of speech samples

## Processing of speech transcripts

The majority of digital recordings of speech were transcribed offline by the first author. A small number were initially transcribed by a research assistant and later checked for accuracy by the first author. A minimal transcription style was adopted. Non-lexical fillers (umm, ah etc.) were not transcribed and pauses were not marked. All lexical items were transcribed, including lexical fillers (e.g., 'like', 'I mean'). Transcripts were submitted to the Stanford Log-linear Part-of-Speech Tagger v3.8 for automated part-of-speech tagging (*Toutanova et al., 2003*). In addition to the coherence measures described in the main text, the following measures were computed for each response:

Speech rate (words per minute): the total number of words produced in the 60s period of each response.

## Proportion closed-class words

The proportion of words in each response whose part-of-speech was classified as closed-class. Closed-class words included pronouns, numbers, prepositions, conjunctions, determiners, auxiliaries and some adverbs.

## Type: token ratio (TTR)

This is the ratio of unique lexical items (types) produced to total words (tokens) spoken. A higher value indicates greater lexical diversity. Because TTR is highly dependent on response length, a moving window approach was adopted to control for the number of words included in the analysis (*Covington and McFall, 2010*). For each response, the first 50 words were first considered and a TTR calculated over these. The window was then moved forward one word and a new TTR calculated, and this process was repeated until the end of the response was reached. A mean TTR was then computed across all windows.

## Mean noun frequency

Frequencies in the SUBTLEX-UK database (*van Heuven et al., 2014*) were obtained for all words tagged as nouns and an average calculated (over tokens) for each response.

## Mean noun concreteness

Concreteness ratings for nouns were obtained from *Brysbaert et al. (2014)*.

## Mean noun age of acquisition (AoA)

Estimates of AoA for nouns were obtained from the norms of *Kuperman et al. (2012)*.

## Mean noun semantic diversity (SemD)

SemD values for nouns were obtained from *Hoffman et al. (2013)*. SemD is a measure of variability in the contextual usage of words. Words with high SemD values are used in a wide variety of contexts and thus more variable and less well-specified meanings.

## Mean noun number of phonemes
The length of all nouns (in phonemes) was also calculated.

## Supplementary LSA methods
The LSA vector space was generated using the British National Corpus and has been described previously (*Hoffman et al., 2013*). Each document in the BNC was divided into contexts of 1000 words in length. A term-by-context matrix was generated, log-entropy weighting applied and singular value decomposition used to reduce the dimensionality of the matrix to 300 dimensions. These steps were performed using the Text to Matrix Generator toolbox in Matlab (*Zeimpekis and Gallopoulos, 2006*) and resulted in the creation of latent semantic vectors for 53,758 words. To generate semantic representations for speech windows or whole responses, the vectors for individual words were combined in the following way:

1. A list of words appearing in the speech window was created. Common function words that carry little semantic information were removed from the list.

2. Vectors for each word in the list were retrieved and normalised so that each had a magnitude of one.

3. The vector for each word was weighted according to (a) the log of its frequency in the speech passage being analysed (assigning greater weight to words occurring more frequently in the window) and (b) its entropy value in the BNC (assigning greater weight to words whose presence is more informative about the semantic content of the window).

4. The weighted vectors were averaged to give a single vector representing the semantic content of the words in the window.

These procedures were similar to those used by other researchers (*Foltz, 2007*; *McNamara et al., 2007*). Examples of high and low coherence responses are shown in Appendix 3-Table 1.

**Appendix 3—table 1.** Examples of high and low coherence responses.

| Example | Prompt | Response |
|---|---|---|
| Low GC (22), low LC (11). Older participant. | Which is your favourite season and why? | My favourite season is spring because it increases the amount of light that you receive from the sun. It's not easy to know why the sun should get bigger but the more you think about it, the more you realise that it's all because in earlier times, people worked out that the earth was a small ball and was affected by the sun, during its travels. In fact, if you're listening to the test match you won't hear anything from Australia unless you get up in the middle of the night and go through to the morning. And that seems to me to, for small boys anyway, is one of the main reasons why it's quite effective to do that. |
| Low GC (26), high LC (37). Young participant. | Describe a typical visit to a restaurant. | I'd get changed. Once I'd got changed, I would get the bus or I would drive in with the family to a restaurant. We'd use the car. We'd drive through into town to get to the restaurant. We'd find a parking space and we'd have a longer walk into town. Once we get into town we'd go through, we'd go up past the bridges looking for a nice place, a place with a view. We'd walk through St. Andrews Square and look for somewhere with a balcony. We'd travel through maybe George Street to somewhere fancy to dine. |

*Appendix 3—table 1 continued on next page*

*Appendix 3—table 1 continued*

| Example | Prompt | Response |
|---------|--------|----------|
| Moderate GC (44), low LC (16). Young participant. | What do people generally do when getting ready for work in the morning? | They usually, you know, get out of bed and stuff and then make their breakfast, and then make their lunch if they're going to be in for lunch as well. And get ready, put their clothes on, and then lock everything up again and leave, however they usually get there. So they might be on the bus or something. Like some people like to have a shower in the morning before work as well. I like to take the recycling out on the way to work, which is good. So they do lots of different things depending on what their work is and depending whether it's in the morning or they're leaving at night time for work. |
| High GC (59), high LC (58). Older participant | Which is your favourite season and why? | My favourite season is spring which, where I live, is quite a dramatic season because there is a tremendous difference from winter, which is cold and everything is, all the flowers and plants appear to be dead, trees have no leaves. So once spring comes, the buds start coming on the trees, plants start bursting through the ground. The earliest ones are probably the snowdrops, followed by crocuses and daffodils and other spring bulbs. And then it moves on, as the spring develops, into the beautiful blossom of cherry and plum and I have a beautiful wild plum tree in my neighbour's back garden, which I enjoy very much every spring. It's my perfect spring tree. The weather generally begins to warm up a bit; we lose any likelihood of getting snow or ice, although it may still be wet and cold. |

DOI: https://doi.org/10.7554/eLife.38907.015

## Appendix 4

DOI: https://doi.org/10.7554/eLife.38907.016

### Validation of coherence measures

To validate the automated LSA-based coherence measures, these were compared with human ratings of GC and LC obtained for a subset of speech samples. Twenty naïve raters were recruited from University of Edinburgh's Psychology department's volunteer panel. The mean age of the raters was 72 and none had taken part in the main experiment. Thirty-two speech samples from two different prompts were selected for rating; each participant rated 16 of these. To collect the ratings, each sample was divided into separate utterances and participants were asked to rate the GC and LC of each utterance on a four-point scale. They were provided with standard definitions of GC and LC (adapted from *Wright et al., 2013*).

Ratings for individual utterances were averaged (weighted by length of utterance) to give mean GC and LC ratings for each speech sample. Ratings from one participant were discarded because they were weakly correlated with the other participants. Following this exclusion, ratings were highly consistent across participants (Cronbach's $\alpha$ = 0.90 for GC and 0.85 for LC). We assessed the relationship between the GC and LC ratings for each speech sample and their corresponding automated measures. There were strong positive correlations in each case ($r$ = 0.68 for GC and $r$ = 0.37 for LC), indicating that the automated measures accurately reflected human coherence judgements. We also tested an alternative method for calculating GC in which a participant's speech sample was compared with a composite vector derived only from the responses of other participants in their own age group. This variation was tested in case there were large differences in the discourse produced by young and older participants which might make between-group comparisons invalid. However, we found that this change had very little effect on GC values or on their correlation with GC ratings ($r$ = 0.71).

Finally, test-retest reliability was assessed in fourteen older participants from the present study, who subsequently took part in a neuroimaging study in which they were asked to produce speech while undergoing fMRI (manuscript in preparation). The speech samples obtained during the fMRI study were transcribed and GC and LC values computed as described above. Participants' mean coherence values from the present study were strongly correlated with the coherence values obtained in the fMRI study (GC: $r$ = 0.88; LC: $r$ = 0.65), indicating a high level of test-retest reliability.

# Appendix 5

DOI: https://doi.org/10.7554/eLife.38907.017

## Additional results

### Semantic task performance

Proportion correct in each condition of the semantic tasks are shown in *Appendix 5—figure 1*. Older adults performed significantly better than young people on the tests of breadth of semantic knowledge (Lexical decision: $t(58) = 8.3$, p<0.001; Synonyms: $t(58) = 9.1$, p<0.001). For the semantic control experiment, accuracy was analysed using a logistic mixed effects model with a $2 \times 2 \times 2$ factorial design. This included age group as a between-subjects factor and task (global vs. feature judgments) and control demands (high vs. low) as within-subject factors. Overall, older adults produced more correct responses than young people ($B = 0.24$, $se = 0.14$, p=0.018). There was a main effect of the semantic control manipulations ($B = -0.72$, $se = 0.13$, p<0.001) and a main effect of task ($B = -0.44$, $se = 0.12$, p<0.001), with poorer performance on the feature selection task. Critically, there was a three-way interaction between group, task and control demands ($B = -0.20$, $se = 0.11$, p=0.042), indicating that age had divergent effects on the two control manipulations.

A separate analysis focused on the feature selection task indicated that there was an interaction between age group and control demands for this task ($B = -0.29$, $se = 0.15$, p=0.037). Older adults showed a larger effect of the manipulation of semantic selection demands (Congruent vs. Incongruent; see *Appendix 5—figure 1*). There was no such interaction for the global association task ($B = 0.11$, $se = 0.14$, p=0.40). These results replicate those previously reported by *Hoffman (2018)*, in a larger sample that included the participants analysed here. They indicate that older adults had particular difficulty selecting among strongly competing active semantic representations. In contrast, controlled retrieval of weak semantic associations did not appear to be impaired in old age.

### Secondary task performance

Error rates in the secondary task were very low, so our analysis focused on RTs. RTs for one participant were not recorded due to a technical issue. The RTs of the remaining participants were log-transformed to reduce skew. On trials where a participant failed to respond within the 2 s time limit (2% of trials), they were assigned the maximum possible RT of 2 s. RTs were entered into a linear mixed effects model that included age group and task as predictors. Estimated means are shown in *Appendix 5—figure 2*. Young people were much faster to respond than older people ($B = 0.27$, $se = 0.026$, p<0.001) and responses were faster in the secondary-only condition ($B = 0.15$, $se = 0.015$, p<0.001). There was, however, no interaction between group and task ($B = -0.002$, $se = 0.010$, p=0.85), indicating that the effects of speech on the secondary task were similar in both groups.

We conducted additional analyses of performance in the dual-task condition that included group and either GC or LC as predictors. This was to test whether participants who responded more quickly in the secondary task produced less coherent speech (i.e., whether this was a trade-off between secondary task performance and coherence). In neither case was coherence a predictor of secondary task RTs (GC: $B = -0.011$, $se = 0.011$, p=0.32; LC: $B = -0.003$, $se = 0.011$, p=0.80).

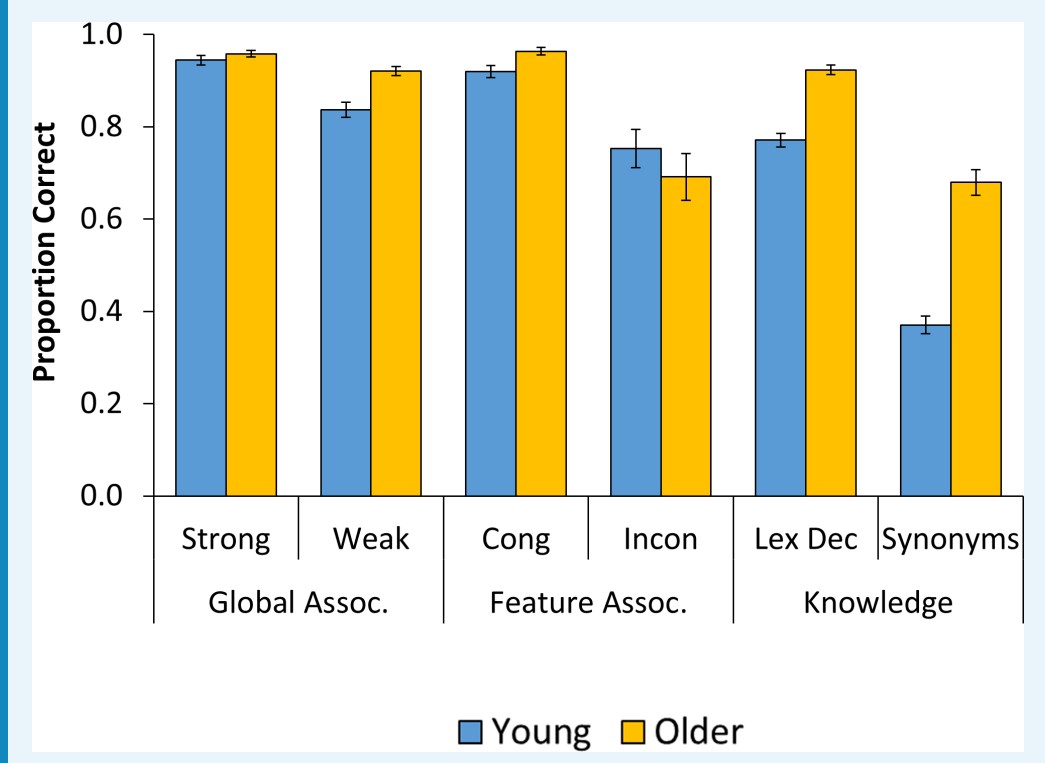

**Appendix 5—figure 1.** Results of tests of semantic processing. Cong = congruent; Incon = incongruent; Lex Dec = lexical decision.

DOI: https://doi.org/10.7554/eLife.38907.018

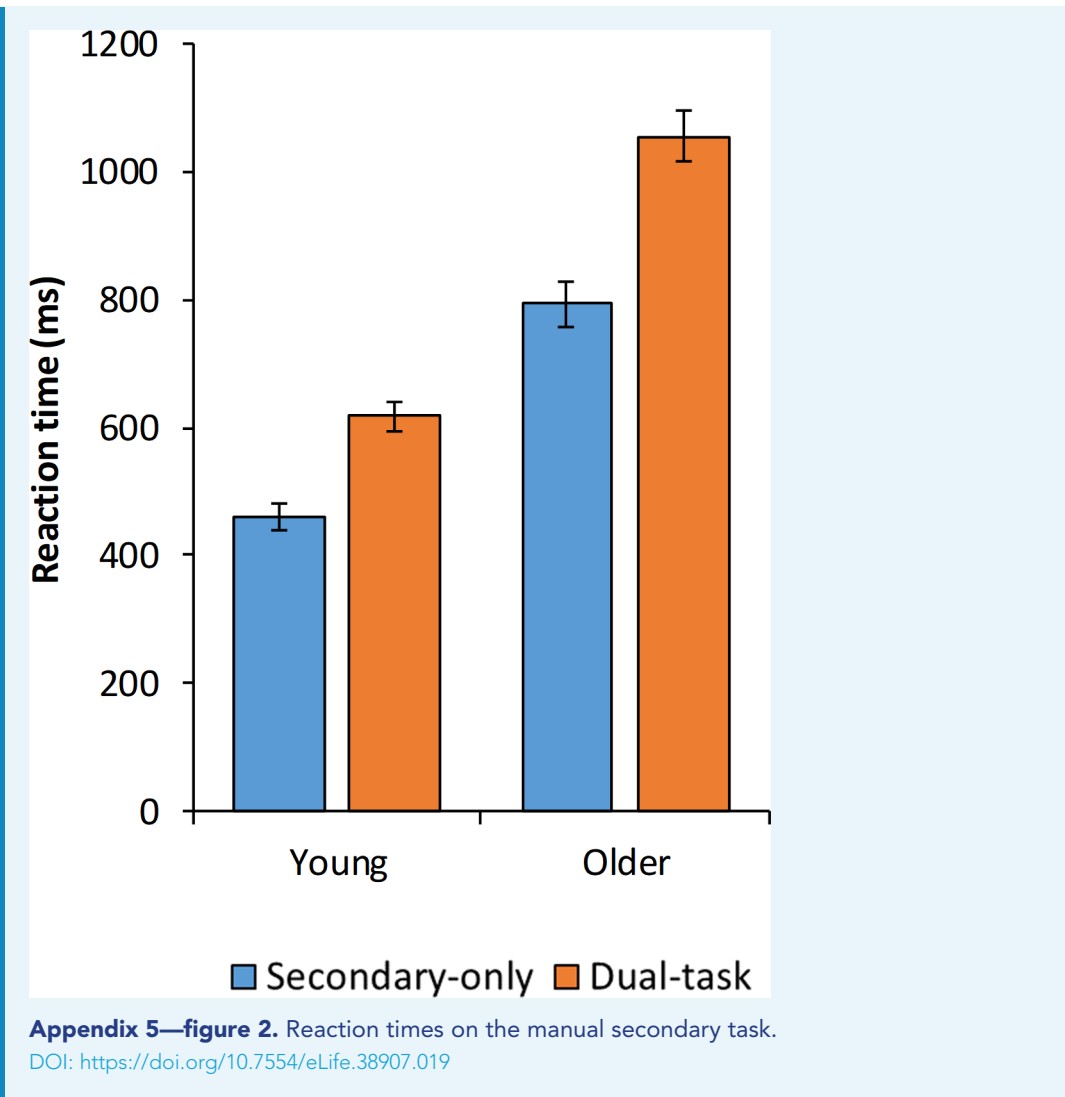

**Appendix 5—figure 2.** Reaction times on the manual secondary task.
DOI: https://doi.org/10.7554/eLife.38907.019

