## [Decision Letter]

Thank you for submitting your article "Poor coherence in older people's speech is explained by impaired semantic and executive processes" for consideration by *eLife*. Your article has been reviewed by two peer reviewers, and the evaluation has been overseen by a Reviewing Editor and Timothy Behrens as the Senior Editor. The following individuals involved in review of your submission have agreed to reveal their identity: Jason Taylor (Reviewer #1); Gary Turner (Reviewer #2).

The reviewers have discussed the reviews with one another and the Reviewing Editor has drafted this decision to help you prepare a revised submission.

Summary:

This manuscript presents a study of the factors underlying age-related changes in speech coherence. An innovative, automated measure of coherence is used, which is based on latent semantic analysis. The study tests the hypothesis that coherence is in part driven by semantic knowledge and semantic selection, not simply domain-general executive function. The main finding is that the semantic cognition measures do indeed predict coherence beyond executive function.

The reviewers agree that this is an impressive investigation that makes a significant contribution to our understanding of the cognitive mechanisms underlying coherent speech production. The rationale and relevance of the study aims are clear and well-grounded in the relevant literature. Study methods and design are both innovative and appropriate to test the hypotheses as presented, and the procedures and results are well described and replicable.

Essential revisions:

The reviewers' evaluations are in broad agreement, as you will see from their individual reviews.

1) Both reviewers felt that the assessment of executive functions was rather limited. Is the Trails B:A ratio score an appropriate way of characterising domain-general executive function? Why was this test used specifically?

To address this concern, you should expand further upon the rationale for this choice of measure, and further highlight the potential implications/limitations of your approach. You may also need to carefully reframe the conclusions to be sure that these are not overstated given this methodological issue. It might help to explicitly acknowledge alternatives that cannot yet be ruled out based on the data. Reviewer 1 also suggests that it might be helpful to report the correlation between semantic selection and executive function, along with the partial correlation.

2) The manuscript should discuss the importance of other cognitive processes which are potentially related behaviourally and neurally to semantic selection. In particular, reviewer 2 points to the large literatures on post-retrieval and working memory. It might not be possible to comprehensively test every alternative account, but discussing these possibilities might increase the relevance of the study to a broader audience whilst also ensuring that the interpretation doesn't extend beyond the data.

*Reviewer #1:*

This manuscript presents a study of the factors underlying age-related changes in speech coherence. An innovative, automated measure of coherence is used, which is based on latent semantic analysis. The study tests the hypothesis that coherence is in part driven by semantic knowledge and semantic selection abilities, and not simply domain-general executive function (the latter association having already been shown elsewhere). The main finding is that the semantic cognition measures do indeed predict coherence beyond executive function, to the extent that age is no longer a significant predictor of coherence when these other variables are included in the model.

The paper is exceptionally clear and (appropriately, given the topic) coherent. The logic, design, and analyses are clear, systematic, and well presented. The results are likely to be of interest to researchers in the fields of ageing and language. This study will fit well with the rapidly growing literature on semantic cognition and its interaction with other abilities.

I have only a few suggestions for how the paper could be improved.

Given the focus on executive function (vs. semantic selection specifically), and given how nebulous – or at least multifaceted – 'executive function' is, the question arises as to whether the Trails B:A ratio is sufficiently representative of domain-general executive function. This choice of task could use further justification.

It is noted that speech rate is a confounder of the LSA-based coherence measure, since the production of more words is likely to drive coherence down. The solution employed in the paper is to include speech rate as a covariate. This is a sound approach, but is it optimal? Couldn't speech rate enter into the calculation of coherence itself, e.g. to normalise the measure (average coherence per word)?

I appreciate the discussion (Discussion, third paragraph) of the potential overlap between semantic selection and other domain-general executive functions. As I read the Results, I couldn't help but wonder whether the Trails ratio measure and semantic selection performance were correlated. I agree that they obviously aren't one and the same, given that they make independent, significant contributions to coherence, but this correlation could inform this discussion and the development of future studies.

I wonder about the validity of the analogy between the adverse effect of semantic knowledge on speech coherence and the 'fan effect' of lower episodic retrieval when multiple pieces of information are associated with the same stimulus (Discussion, fifth paragraph). The former is about breadth (more information overall), whereas the latter is about depth (more information associated with a specific thing. This (perceived, on my part at least) lack of correspondence isn't necessarily damaging to the overall interpretation of the effect, but it may confuse the issue in some readers' minds.

This may expose my naivety regarding linear mixed modelling, but is Chi square the best measure of improvement of fit between models? Does it adequately penalise complexity?

*Reviewer #2:*

The authors set out to investigate the contributions of semantic cognition to speech coherence in younger and older adults. They employ innovative computational methods to derive measures of global and local coherence. Further, they included a measure of executive functioning (Trails B:A) as well as a dual-task condition to assess the contribution of semantics over and above cognitive control and attentional demand to age differences in speech coherence.

Overall, this is an impressive investigation, one that makes a significant contribution to our understanding of the cognitive mechanisms underlying an important age-related shift in functional capacity (i.e. coherent speech production). The rationale and relevance of the study aims are convincingly argued and appear to be well-grounded in the relevant literature. The hypotheses reflect, and follow logically from previous findings. Study methods and design are both innovative and appropriate to test the hypotheses as presented, and the procedures are well described and replicable. The presentation of the results is clear and comprehensive and the authors' interpretations and conclusions are fully supported by the data.

I have only a couple of issues, both of which were in part already identified by the authors, although perhaps more prominence/discussion is warranted. The first is a conceptual issue surrounding the authors' interpretation of their findings as evidence for the contribution of semantic ability (and semantic selection more specifically) in speech coherence. As discussed in the second paragraph of the Discussion, semantic selection closely overlaps (both behaviourally and neurally) post-retrieval monitoring processes, an area with a rich literature in the context of neurocognitive aging (although a quick literature search revealed only tangential references to speech production specifically). While the authors provide a convincing case for semantic selection based on their targeted measures, it is unclear whether the authors would argue that this is distinct from, or possibly a domain-specific instantiation of, post-retrieval monitoring. Some elaboration of this point seems warranted to further contextualize the findings and draw relevant links to the broader monitoring literature. On a related point, neuropsychologically, repetition errors on the fluency tasks are often used to assess monitoring. While likely an insensitive metric in a typically aging cohort, presenting these data may provide some insight with respect to age differences in monitoring more broadly defined.

The second concern involves the narrow definition of executive control processes, operationalized here by a single measure. While the authors do acknowledge this limitation, it again would seem to warrant further discussion with respect to the potential implications of untested interactions between other executive control processes (e.g. working memory capacity) and semantic processes. It seems at least plausible that a more comprehensive measure/index of executive control might be a stronger predictor of speech coherence than the semantic/Trails model tested here. It is certainly not incumbent on the authors to test every alternative hypothesis. However, the central argument of the paper (as indicated in the title) is that *both* executive control and semantic processes contribute to reduced coherent speech production in older adults, yet the executive control side of that argument has not been comprehensively assayed here. Further, it is possible that semantic abilities may come to play a larger role in the context of declining control processes in later life, something that we've demonstrated both behaviorally and neurally in the context of autobiographical memory in recent work (Spreng et al., Neuropsychologia, 2018). As with the first point above, a discussion of these possibilities would help to enhance the scope/impact of findings beyond speech coherence with implications for neurocognitive aging more broadly.

---

## [Author Response]

Essential revisions:The reviewers' evaluations are in broad agreement, as you will see from their individual reviews.1) Both reviewers felt that the assessment of executive functions was rather limited. Is the Trails B:A ratio score an appropriate way of characterising domain-general executive function? Why was this test used specifically?To address this concern, you should expand further upon the rationale for this choice of measure, and further highlight the potential implications/limitations of your approach. You may also need to carefully reframe the conclusions to be sure that these are not overstated given this methodological issue. It might help to explicitly acknowledge alternatives that cannot yet be ruled out based on the data. Reviewer 1 also suggests that it might be helpful to report the correlation between semantic selection and executive function, along with the partial correlation.

We agree that our assessment of domain-general executive function was rather limited. Unfortunately, we were unable to run a more extensive battery of tests within the time available to test each participant. We selected the Trails test as our executive measure because previous studies have found correlations between this test and coherence in speech (Arbuckle and Gold, 1993 and Wright et al., 2014 – as noted in the Introduction) and because it did not require lexical processing (unlike, say, the Stroop task). We used the ratio of B to A completion times as our dependent measure because previous studies have indicated that this measure is largely independent of general processing speed, which shows steep declines with age (Arbuthnott and Frank, 2000; Salthouse, 2011 – as noted in the Materials and methods).

We have added the following text to the Discussion to consider the implications of our choice of executive function task:

“In this study we employed a single measure of non-semantic executive ability, the Trails test. […] This is an important issue to resolve if we are to understand how semantic selection processes relate to domain-general executive function, particularly as semantic selection deficits in later life may be related to more general declines in inhibitory function, which have been reported across a range of tasks (Borella, Carretti, and De Beni, 2008; Hasher and Zacks, 1988; Hoffman, 2018; Salthouse and Meinz, 1995).”

2) The manuscript should discuss the importance of other cognitive processes which are potentially related behaviourally and neurally to semantic selection. In particular, reviewer 2 points to the large literatures on post-retrieval and working memory. It might not be possible to comprehensively test every alternative account, but discussing these possibilities might increase the relevance of the study to a broader audience whilst also ensuring that the interpretation doesn't extend beyond the data.

We agree that we should have provided a more detailed consideration of how semantic selection links with other related cognitive processes, and in particular with the literature on post-retrieval processing of episodic memories. We have added the following passage to the Discussion:

“A large literature has examined brain regions implicated in the selection of task-relevant aspects of retrieved episodic memories. These processes are often referred to as “post-retrieval monitoring”. […] We have demonstrated here that this selection deficit contributes to the loss of coherence in later life.”

Reviewer #1:[…] I have only a few suggestions for how the paper could be improved.Given the focus on executive function (vs. semantic selection specifically), and given how nebulous – or at least multifaceted – 'executive function' is, the question arises as to whether the Trails B:A ratio is sufficiently representative of domain-general executive function. This choice of task could use further justification.

Please see response to Essential revisions 1.

It is noted that speech rate is a confounder of the LSA-based coherence measure, since the production of more words is likely to drive coherence down. The solution employed in the paper is to include speech rate as a covariate. This is a sound approach, but is it optimal? Couldn't speech rate enter into the calculation of coherence itself, e.g. to normalise the measure (average coherence per word)?

This is certainly a possibility. We did not take this approach in the study because we did not know a priori whether speech rate would be related to coherence. Rather than make this assumption, we decided to include speech rate as a covariate in the model so that the effect on coherence would be tested empirically. Another reason for preferring this approach is that our two coherence measures have different relationships with speech rate: speech rate predicted global coherence but not local coherence, so it is not clear whether one should “correct” one measure but not the other.

I appreciate the discussion (Discussion, third paragraph) of the potential overlap between semantic selection and other domain-general executive functions. As I read the Results, I couldn't help but wonder whether the Trails ratio measure and semantic selection performance were correlated. I agree that they obviously aren't one and the same, given that they make independent, significant contributions to coherence, but this correlation could inform this discussion and the development of future studies.I wonder about the validity of the analogy between the adverse effect of semantic knowledge on speech coherence and the 'fan effect' of lower episodic retrieval when multiple pieces of information are associated with the same stimulus (Discussion, fifth paragraph). The former is about breadth (more information overall), whereas the latter is about depth (more information associated with a specific thing. This (perceived, on my part at least) lack of correspondence isn't necessarily damaging to the overall interpretation of the effect, but it may confuse the issue in some readers' minds.

We have removed the reference to this effect to avoid any confusion.

This may expose my naivety regarding linear mixed modelling, but is Chi square the best measure of improvement of fit between models? Does it adequately penalise complexity?

The likelihood ratio test (with a chi-square statistic) is widely used for assessing change in model fit when an effect is added to a linear mixed effects model. Although other approaches are available, simulation data indicate that this method is most effective for minimising Type I errors (Barr et al., 2008). As stated in the Materials and methods, we followed Barr et al.’s recommendations for constructing our mixed effects models, including the use of the likelihood ratio tests to assess the significance of fixed effects.

Reviewer #2:[…] I have only a couple of issues, both of which were in part already identified by the authors, although perhaps more prominence/discussion is warranted. The first is a conceptual issue surrounding the authors' interpretation of their findings as evidence for the contribution of semantic ability (and semantic selection more specifically) in speech coherence. As discussed in the second paragraph of the Discussion, semantic selection closely overlaps (both behaviourally and neurally) post-retrieval monitoring processes, an area with a rich literature in the context of neurocognitive aging (although a quick literature search revealed only tangential references to speech production specifically). While the authors provide a convincing case for semantic selection based on their targeted measures, it is unclear whether the authors would argue that this is distinct from, or possibly a domain-specific instantiation of, post-retrieval monitoring. Some elaboration of this point seems warranted to further contextualize the findings and draw relevant links to the broader monitoring literature. On a related point, neuropsychologically, repetition errors on the fluency tasks are often used to assess monitoring. While likely an insensitive metric in a typically aging cohort, presenting these data may provide some insight with respect to age differences in monitoring more broadly defined.

Please see response to Essential revisions 2.

*The second concern involves the narrow definition of executive control processes, operationalized here by a single measure. While the authors do acknowledge this limitation, it again would seem to warrant further discussion with respect to the potential implications of untested interactions between other executive control processes (e.g. working memory capacity) and semantic processes. It seems at least plausible that a more comprehensive measure/index of executive control might be a stronger predictor of speech coherence than the semantic/Trails model tested here. It is certainly not incumbent on the authors to test every alternative hypothesis. However, the central argument of the paper (as indicated in the title) is that* both *executive control and semantic processes contribute to reduced coherent speech production in older adults, yet the executive control side of that argument has not been comprehensively assayed here. Further, it is possible that semantic abilities may come to play a larger role in the context of declining control processes in later life, something that we've demonstrated both behaviorally and neurally in the context of autobiographical memory in recent work (Spreng et al., Neuropsychologia, 2018). As with the first point above, a discussion of these possibilities would help to enhance the scope/impact of findings beyond speech coherence with implications for neurocognitive aging more broadly.*

Please see response to Essential revisions 1.